# The Diversity of *Wolbachia* across the Turtle Ants (Formicidae: *Cephalotes* spp.)

**DOI:** 10.3390/biology13020121

**Published:** 2024-02-13

**Authors:** Corey Reese, Leland C. Graber, Manuela O. Ramalho, Corrie S. Moreau

**Affiliations:** 1Department of Entomology, Cornell University, Ithaca, NY 14853, USA; cmr282@cornell.edu (C.R.); corrie.moreau@cornell.edu (C.S.M.); 2Department of Biology, West Chester University, West Chester, PA 19383, USA; mramalho@wcupa.edu; 3Department of Ecology and Evolutionary Biology, Cornell University, Ithaca, NY 14853, USA

**Keywords:** ants, *Wolbachia*, multi locus sequence typing, biogeography

## Abstract

**Simple Summary:**

One of the most widespread bacteria associated with insects are those in the genus *Wolbachia*. These bacteria can cause a wide range of behavioral and physiological changes within their host species, but not all these interactions are well-documented. Specifically in ants, the diversity of *Wolbachia* has not been characterized fully leaving many open questions regarding its interactions with this globally distributed insect family. This study surveyed the genus *Cephalotes* for its *Wolbachia* diversity and aimed to characterize the singly infected samples found along with discovering any patterns in infections. Using multilocus sequence typing, novel genes and *Wolbachia* strains were discovered across the native distribution of *Cephalotes*. There were no clear patterns in infections nor any association with biogeographic distribution. Overall, the unique sequences should be investigated further but provide important information on the distribution and relationship between *Cephalotes* and *Wolbachia* at the genus level. This provides further information for future studies to continue investigating these unusual bacteria and their complex relationships with ant-hosts.

**Abstract:**

*Wolbachia* is a widespread and well-known bacterium that can induce a wide range of changes within its host. Ants specifically harbor a great deal of *Wolbachia* diversity and are useful systems to study endosymbiosis. The turtle ants (*Cephalotes*) are a widespread group of tropical ants that rely on gut microbes to support their herbivorous diet for their survival, yet little is known of the extent of this diversity. Therefore, studying their endosymbionts and categorizing the diversity of bacteria within *Cephalotes* hosts could help to delimit species and identify new strains and can help lead to a further understanding of how the microbiome leads to survival and speciation in the wild. In our study, 116 individual samples were initially tested for positive infection with the *wsp* gene. Of the initial 116 samples, 9 samples were infected with only one strain of *Wolbachia*, and 7 were able to be used successfully for multilocus sequence typing (MLST). We used the new MLST data to infer a phylogeny with other Formicidae samples from the MLST online database to identify new *Wolbachia* strains and related genes, of which only one came back as an exact match. The 18 *Wolbachia*-positive samples ranged across 15 different species and 7 different countries, which we further test for species identity and geographic correlation. This study is the first comprehensive look into the diversity of *Wolbachia* in the turtle ants, providing insight into how endosymbionts are oriented in widespread species and providing a strong foundation for further research in host-microbe interactions.

## 1. Introduction

Ants have developed and rely on complex interactions with their internal microbial symbionts and the related biochemical systems of microbes for their own development, such as cuticle formation or the egg laying of the queen [1,2]. While some symbionts have helped contribute to the overall success of ants, many of the specific consequences and functions associated with microbes are relatively unknown [3]. This is the case of the bacteria *Wolbachia*, which is known to alter the behavior or limit the reproductive success of many of its hosts. It can induce cytoplasmic incompatibility, parthenogenesis, male killing, and feminization [4,5]. However, it also can help protect its host from viruses and aid in nutrient acquisition. Further understanding the biology of *Wolbachia* and their host-associated diversity has important applications in biological control for disease or pest outbreaks [5,6]. The breadth of the groups of insects and other arthropods the bacterium infects still requires much study, especially in ants [3,7]. Studies have documented the spread of *Wolbachia* among invasive species [8] and the diversity among native ranges of *Polyrhachis* spp. [9] and *Cephalotes atratus* [10] ant hosts. A focus of many studies has also been understanding the horizontal and vertical transmission possibilities of *Wolbachia* transfer among ants. It is speculated that infections are often transmitted vertically and less frequently horizontally only between closely related species [11].

*Wolbachia* has been understood as an infectious insect bacterium since the early 1900s from studies on *Culex pipiens* [12,13]. The bacterium is transmitted vertically between species and can persist as multiple infections within its host. This was discovered through observed cytoplasmic incompatibility and continued to be seen through further studies [14,15,16]. To molecularly identify *Wolbachia*, an earlier method was to look at the *Wolbachia* surface protein (wsp), which is highly conserved across species, serving as a good indicator of infection [17]. While this provides initial screening success, it undergoes a wide range of recombination between *Wolbachia* clades and makes it an unreliable method for understanding *Wolbachia* diversity. Hence, the multilocus sequencing typing (MLST) method was developed by Baldo et al. [18], utilizing the five genes *coxA*, *ftsZ*, *fbpA*, *gatB*, and *hcpA* to develop a unique sequence for each sample.

MLST has provided a more robust approach to study *Wolbachia* and has been used for a wide range of host organisms to develop a library of available sequences with which to compare future data [17]. This includes a database of bacterial and host information (https://pubmlst.org/organisms/wolbachia-spp, accessed on 4 October 2023). The database has over 2000 allele entries for comparison and identification of sequences. This catalog of *Wolbachia* helps to organize the wide diversity of sequences, hosts, and geographic sampling locations, most of which are detailed in each MLST submission. The identification of *Wolbachia* can have variable results, as this information is organized into twenty supergroups ranging from A to U [19,20]. The supergroups are divided based on phylogenies created from *16S rDNA*, *wsp*, and *ftsZ* markers [20,21]. Of these supergroups, the most common ones found in ants are A and F [19]. Ramalho and Moreau [22] suggested that the origin of *Wolbachia* in ants occurred in Asia [22]. This resulted in supergroup F spreading to Asia and Africa, with minimal associations with ants. Ants have a stronger relationship with supergroup A, which is associated with movement into North and South America. Modern *Wolbachia* strains seem to be grouped according to New and Old-World distributions, and this pattern has been observed across multiple studies [9,11,22]. This information provides a solid foundation to continue looking at infection rates across species within Formicidae [22].

*Cephalotes* is an ant genus of 123 species located in the Americas, with the majority of species diversity found in Central and South America [23,24,25]. They are known as turtle ants due to the flat head that the soldier caste possesses, which is used to block the opening of the nest entrance. Nests can often be found in hollowed-out stems where the soldiers can take advantage of small holes to use their heads as blockades [26,27]. An unusual quality of turtle ants is that they are herbivorous, a rare diet for ants. Many species forage in the canopy and have been noted to supplement their diet with mammal urine and bird feces to acquire nitrogen [28]. Furthermore, they have a well-developed gut microbiome to help them recycle urea to preserve nitrogen in their bodies, which is essential for many natural processes [28]. This diversity of microbes and unusual natural history makes *Cephalotes* an informative system to study host relationships with *Wolbachia* [28].

Additionally, the host’s environment has been shown to influence the bacterial symbiont diversity within a host species [10,29,30]. Specifically, *Cephalotes* ants have been seen to have variations in their bacterial communities depending on geographic distribution [31]. Thus, testing for associations between symbiont diversity and biogeography has become an important aspect of analyzing *Wolbachia* communities. Biogeography has been explored as a factor that could impact *Wolbachia* diversity within studies on *Polyrhachis* spp. and *C. atratus* [9,10]. While no significant relationship was observed between the bacterium and *Polyrhachis* species, there was a significant correlation between geographic location and *Wolbachia* in *C. atratus* [9,10]. This information, in combination with the family-wide analysis of *Wolbachia* in Formicidae, provides a solid foundation to continue looking at infection rates across species [22]. Our study is meant to further test the idea that strains are correlated with biogeographical distribution while observing genus-wide infection rates. 

In this study, we analyzed the infection rate and pattern of distribution for *Wolbachia* in the genus *Cephalotes*; our primary objective was to determine the diversity of *Wolbachia* in *Cephalotes* ants and to observe any potential patterns in infection across samples. We used sequence typing to answer the following questions: (1) Does geographic location impact *Wolbachia* diversity in *Cephalotes* spp.? (2) What is the infection rate for a sample size of *Cephalotes* populations across their native distribution? (3) Are there any patterns present in the species that confer infections? (4) Finally, what are the sequence types of the sampled *Wolbachia*? Given each of these questions, we predict statistically significant associations between *Wolbachia* infections and *Cephalotes* spp. phylogenetic positions and biogeographic distribution.

## 2. Methods

A total of 116 *Cephalotes* samples from across seven countries were tested for the presence of *Wolbachia* infection. These samples represent 15 species with a total of eighteen positive samples (Figure 1). The specific details of the *Wolbachia*-positive samples can be found in Table 1. 

The complete list of tested samples can be found in the Appendix A. After collection in the field, specimens were immediately stored in 95% ethanol and kept at −20 °C until DNA extraction. The DNA extractions were carried out through a DNeasy Blood and Tissue protocol from Qiagen utilizing whole-body ant specimens. The samples were stored, again, at −20 °C. The geographic range and breadth of species in the sampled group was meant to be representative of the *Cephalotes* genus distribution. The *wsp* gene was used to determine which samples were positive for *Wolbachia* following PCR amplification. The *wsp* gene was amplified using Taq DNA Polymerase and the wsp81f and wsp69r primers (each at a concentration of 1 µM) followed by the addition of 1 µL of DNA [28]. The process was 36 cycles long with a 59 °C annealing temperature. The following thermal cycling sequence was composed of an initial denaturation at 94 °C for 30 s, followed by an annealing time for 45 s, extension for 90 s at 72 °C, a final elongation step at 70 °C for 10 min, and a hold at 4 °C. The annealing temperature for each of the genes varied and were as follows: 52 °C for *ftsZ*, 54 °C for *gatB*, 53 °C for *hcpA*, and 55 °C for *coxA* and *fbpA*. Gel electrophoresis was used to analyze PCR products through a 1% agarose gel. The presence of a band in the gel was noted as at least one infection present in the sample. Positive *Wolbachia* PCR products were purified using Exosap (Cleveland, OH, USA) and the specified thermocycler settings the company recommends. The purified samples then underwent Sanger sequencing and were prepared using BigDye Terminator from Applied Biosystems (Waltham, MA, USA). The sequencing was performed through the Cornell Institute of Biotechnology (Ithaca, NY, USA). The program Geneious Prime 2023.1 (https://www.geneious.com, accessed on 10 August 2023) was used to analyze the electropherograms and determine whether samples were infected with a single or multiple strains of *Wolbachia*.

A subset of the *Cephalotes* samples, i.e., those with single *Wolbachia* infections (*n* = 9), went through MLST analysis. Five amplified MLST genes were then sequenced (*coxA*, *fbpA*, *ftsZ*, *gatB*, and *hcpA*) following a protocol that was similar to that of the *wsp* gene. The other group of positive samples, i.e., those infected with multiple strains of *Wolbachia*, was excluded due to the difficulties that arise when trying to decipher overlaid Sanger sequences; this exclusion is consistent with other similar *Wolbachia* studies [9,10,32]. All of the sequence alignments were generated using Geneious Prime 2023.1 and were compared to reference sequences in the MLST online database. The sequences for the five MLST genes (*coxA*, *fbpA*, *ftsZ*, *gatB*, and *hcpA*) from the nine positive samples were trimmed and concatenated, totaling 2103 base pairs in length. The sequences were combined with 90 MLST sequences from other *Wolbachia* strains found with Formicidae. All of the data were inserted into IQ-Tree web server version 2.2.2.7 [33] and used to create a maximum likelihood phylogenetic inference with one thousand bootstrap replicates to measure support of the tree topology. ModelFinder via IQ-Tree was used to find the best-fitting substitution model for each locus; partitions and substitution models specified for each are included in the Appendix A. Samples of *Wolbachia* from *Ocymyrmex picardi* and *Paratrechina* sp., namely ST124 and ST557, respectively, were used to form the outgroup of the phylogenetic tree. Both of these samples belong to supergroup F. 

The R package vegan was used to conduct two Mantel tests [34]. The two tests measured the correlation between a distance matrix constructed from a pruned *Wolbachia* phylogeny featuring only the *Cephalotes* specimens and two distance matrices: (1) a distance matrix created from a newly inferred UCE *Cephalotes* phylogeny [28,35], pruned to only include species corresponding to the *Cephalotes* ant hosts in the *Wolbachia* phylogeny, and (2) a distance matrix created from the latitude and longitude values at which each *Cephalotes* ant was collected.

## 3. Results

Of the total 116 *Cephlaotes* samples tested, 18 tested positive (15.5%) after being tested for *wsp*. From those positive samples, nine had a single *Wolbachia* infection, and the other nine had multiple infections. The area samples ranged from the southern United States to Argentina, encompassing 19 countries. Of the total countries sampled, nine had at least one reported positive sample. The singly infected ants were collected from five countries, and the multiply infected ants were collected from four countries (Figure 2). A total of 57 species were tested, of which 15 species included positive samples. One sample did not have a recorded species associated with it. Of the 15 species with positive samples, two species had both single- and multiple-infected samples collected, with these two species being *C. clypeatus* and *C. umbraculatus* (Figure 3). A total of 17 samples were able to be linked with a specific species, while C18_32 was the only positive sample that was identified only to the *C. ngustus* clade. 

Of the nine singly infected samples, only one produced sequences for all five MLST genes. The reason for this high sequencing failure is unknown. However, four of the other samples produced sequences for four of the five MLST genes, while the other three sequenced samples had varying ranges of unsequenced genes (Table 2). To obtain these allele numbers, MLST sequences were blasted against the PubMLST database to find closely related alleles, resulting in the number seen in Table 2. We were unable to obtain any sequences from one sample, so it was excluded from the MLST gene generation, but we could confirm that it was singly infected from the *wsp* screening. This sample was CSRA_1185, which was *C. cristatus* collected from San José, Costa Rica. The differences between MLST sequences is further illustrated in a haplotype network (Figure 4).

Only one of the singly infected samples yielded a full sequence type. All of the other samples had at least one of the five MLST genes missing. One of the samples, CSRA_1185 was unable to produce a viable sequence for any of the genes when we tested allele sequence type against the database. 

The phylogeny (Figure 5) was inferred with 98 *Wolbachia* MLST sequences, with 8 of them being the new *Cephalotes* additions. Since none of the sequences are associated with the outgroup, the new *Wolbachia* strains are within supergroup A. None of the species that were inferred in the phylogeny were represented multiple times other than *C. atratus*, which was already present in the database before the addition. The new *C. atratus* sample did not form within the clade of the pre-existing *C. atratus* data. All of the other species were new additions to the phylogeny. Furthermore, the haplotype map of each gene shows how samples differed from one another (Figure 4).

Results of the Mantel tests determining if there was a correlation between geographic location and a *Wolbachia*-infected sample found no significant correlation between the *Cephalotes*-hosted *Wolbachia* phylogenetic distance matrix and either of the two tested matrices. The Mantel statistic and *p*-value of the test with the *Cephalotes* phylogeny were 0.0908 and 0.3298, respectively; the Mantel statistic and *p*-value of the test with the latitude–longitude distance matrix were −0.1177 and 0.8282. 

## 4. Discussion

This study documents the distribution of *Cephalotes* samples to correlate the geographic spread of infections with subsequent *Wolbachia* diversity, which allows for a better understanding of ant host–microbe interactions. To add to this information, we determined the overall infection rate looking at a genus-wide survey of these ants with analysis to observe any related patterns of infection. Finally, we determined the uniqueness of the sequence types in the positive samples, encompassing the objectives of the study. 

The sampled ants spanned most of the native range of *Cephalotes*, giving us a broad basis in which to look for an association between infection and host samples. Of the 19 countries’ samples, seven had at least one positive sample (37%). The sampling was not even across geography (Figure 1). While countries do not provide the same information as habitats, this form of analysis is used to represent how the infections span across their native range, which extends across the Americas. Increased sampling could find different relationships between biogeography and *Cephalotes* infections, as this has been found in other studies only looking at the species *C. atratus* [10]. However, other studies have shown that a link between biogeographic distribution and the sequence type of the bacteria is unlikely, as seen in groups like the ants *Polyrhachis* and various termite species [9,36]. Our study found a similar result, as the Mantel tests did not indicate that host biogeography and phylogeny have a significant impact on the infection status of a *Cephalotes* ant. 

Our analysis shows a 15.5% infection rate across the *Cephalotes* species. Using the information from Figure 3, 26.3% of species showed at least one infection out of the 57 sampled. This rate is lower when compared to other studies looking at *Wolbachia* infection rates among ants. The genus *Polyrhachis* was found to have a 47% infection rate when looking at individual species [9]. Infection rates among *Solenopsis* colonies reached 51% [37]. A survey of ant species in southern China found rates of *Wolbachia* infection around 35.71% among 56 species [38]. While some of these studies measured infection rate in different ways, the overall trend shows higher infection rates in various other ant groups than what was observed in *Cephalotes*. It is unclear why this particularly low rate was observed, but it is an avenue for further research.

In the phylogeny, the *Cephalotes*-associated samples were distributed throughout multiple different clades. The nonsignificant result of the geographic distance Mantel test can be seen further in the makeup of the phylogeny, as the geographic distribution of the samples spans across multiple regions, and there is no apparent pattern of relatedness. Some of the newly added samples from this study were inferred to be in clades alongside *Cephalotes* samples already included in the phylogeny, such as the *coxA* sequences from POW0460A and C18_32. The other samples were inferred to be in a variety of different clades, showing the potentially wide biodiversity of *Wolbachia* within ant genera and collection locations. For example, the newly sequenced *C. atratus* sample (CSM2997) did not appear in the clade of the pre-existing *C. atratus* data, indicating a large potential for diversity even in the same species. A similar spectrum of diversity has been seen in other studies where infected samples relayed a similar variety of sequence types [11].

The sequencing typing of the samples as well as the haplotype networks indicate that the sequences are largely unique when compared with other previous *Wolbachia* MLST sequences. The *gatB* sequence of POW471 was the only one that had an exact match of its sequence, and it seems to be closely related to other previously existing samples within the phylogeny. Otherwise, samples seem to group together in small clades and form new branches within them, which was seen with C18_307 and CSM3433. The newly added samples were placed within this association of *C. atratus* at the base of the phylogeny. These samples, POW0460A and C18_32, were *C. liepini* and an unspecified species in the *Angustus* clade, respectively. Neither of these, regardless of the species in the *Angustus* clade, phylogenetically place close to *C. atratus* [35]. Therefore, it is likely that species-specific infections are not occurring, and biogeographic association could thus only occur for select groups of *Cephalotes* communities. 

Since nearly all of the MLST genes did not have exact matches to previously sequenced hosts, it could be possible that the varied niches occupied by the *Cephalotes* hosts contributed to this. It has been shown that the ecological niche and species group of an organism can influence microbial symbionts of a host species [32,39]. This could confer specific evolutionary and ecological relationships that select for specific *Wolbachia* sequence types that may not be *Cephalotes*-spp.-specific but provide novelty for their life history, with similar traits found in other samples. This is because there are non-*Cephalotes* species that lay within the new clades formed from these samples, but the majority of the clades are composed of the newly added sequence types as well. Thus, there could be some overlap between the diversity of *Wolbachia* and the conferred relationship beyond the niche seen in the turtle ants. Similar patterns are seen in other taxa as well. Samples derived from *Polyrhachis* and *Solenopsis* fall into clades where they closely associate with other samples of the same genus; however, this is not true across all samples within the phylogenies [9,40]. It should also be noted that missing MLST data could contribute to error in phylogenetic and sequence type analysis; two of our samples only had one gene sequenced. However, even the most completely sequenced of our samples did not appear closely related to the *Wolbachia* from *Cephalotes atratus* samples sequenced by Kelly et al. in 2019 [10], so we cannot establish a clear link between the missing data and their impact on phylogenetic accuracy. 

Since all of the *Cephalotes* samples are nested within the major clade, they all fall under the supergroup A classification. As the samples spanned nearly the entire native range of these ants, it is likely that the associated *Wolbachia* strains that infect them fall solely within supergroup A. Strains associated with Formicidae often associate with supergroup A and B, but the majority fall under supergoup A [8,11,22]. Not every region of their habitat was included; however, the expanse of sampling completed likely covers enough breadth and diversity within their native region to draw conclusions about these patterns.

## 5. Conclusions

This study highlights the importance of surveying specific groups of organisms for their microbial relationships, such as with the *Wolbachia* diversity in *Cephalotes*. This is especially important for groups of organisms that inhabit unique niches in the environment, which could confer equally unique strains of their microbial symbionts. The ways that microbes influence the evolution and ecology of ants are only partially understood but have large implications for grasping the full image of these interactions. Future studies should continue to investigate *Wolbachia* for its diversity in not only ants but in other orders of insects, too. Specifically, with *Cephalotes*, studies should continue to look at the *Wolbachia* diversity across its native range, as it seems to harbor a wide diversity of novel strains and could represent unusual relationships between ants and the symbionts. Bringing more sampling to other studies could bring a combination of data together to fully understand how *Wolbachia* associates and influences the lives of this unusual genus of ants. Additionally, understanding generally how host and microbes associations take place could lead to elucidating the function and importance of host–microbe interactions.

## Figures and Tables

**Figure 1 biology-13-00121-f001:**
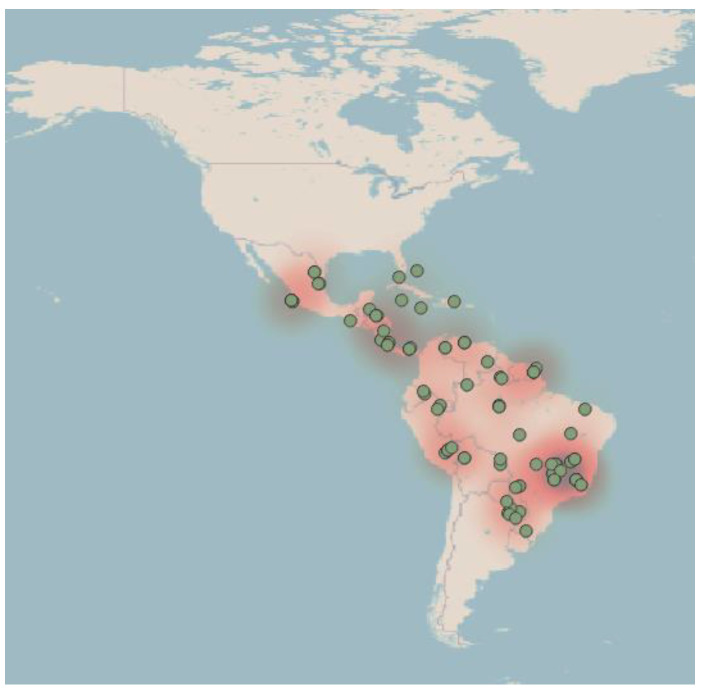
A heatmap showing where all of the samples were taken for this study and the relative quantity of samples analyzed from each area. Some locations are more represented than others, but the sample range covers a wide distribution of habitat. Each green dot represents the location of a sample used analysis.

**Figure 2 biology-13-00121-f002:**
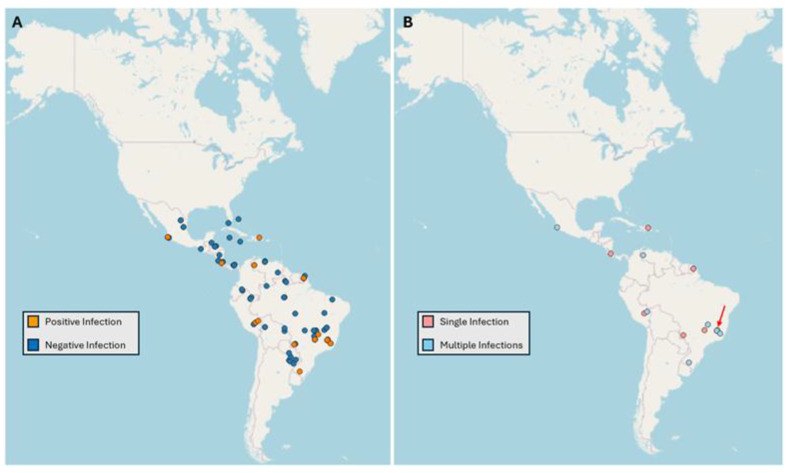
(**A**) The geographic distribution of all of the samples that were tested. (**B**) The distribution of the samples that tested positive for *Wolbachia* infections organized by their infection status. The red arrow indicates a point of three multiply infected samples that are close in geographic location and cannot be differentiated based on the spacing of the map.

**Figure 3 biology-13-00121-f003:**
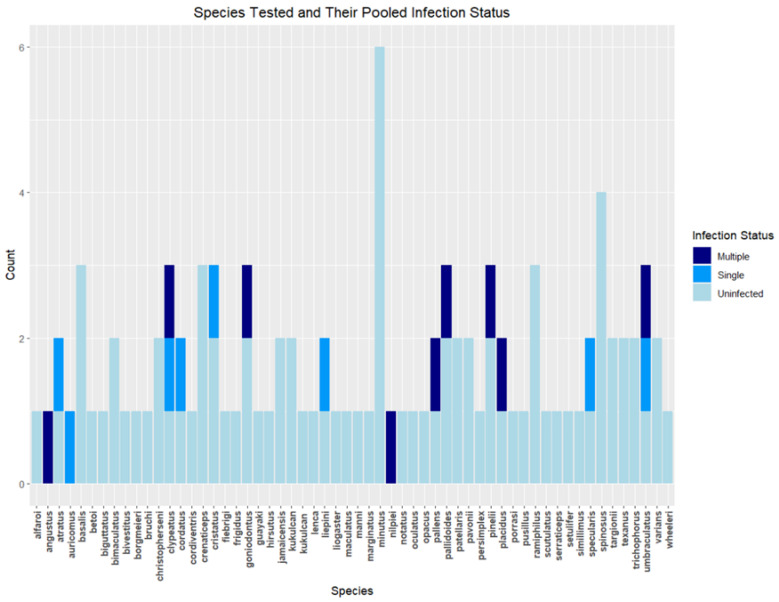
A figure representing the diversity of species sampled with their respective infection demographic shown. Of the 57 species, 15 showed at least one positive sample, and 2 had both single- and multiple-infected samples. There were 17 total positive samples that were associated with species, and 1 sample was unidentifiable beyond the clade level.

**Figure 4 biology-13-00121-f004:**
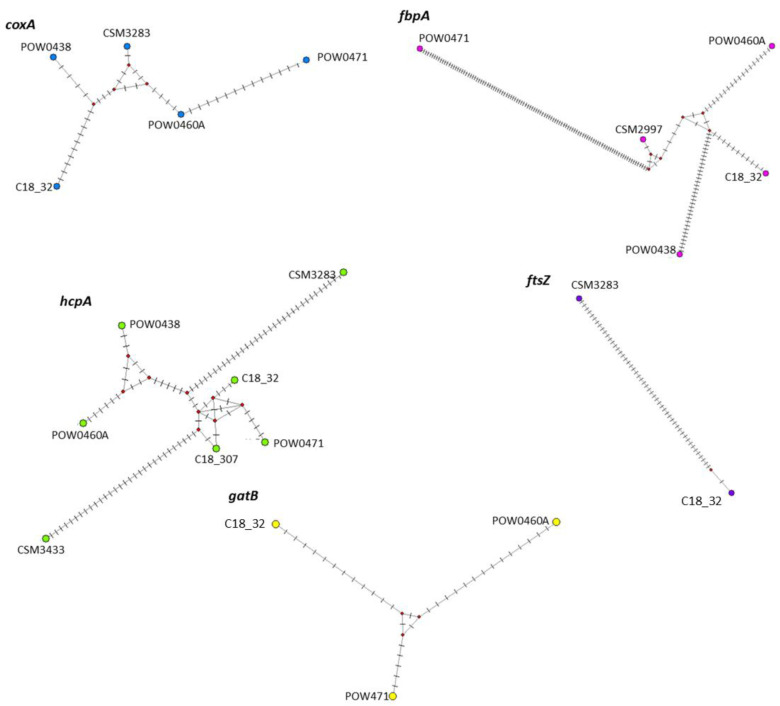
Haplotype network figures showing each sample’s sequence for all MLST genes. Different genes are symbolized by different colored dots. Black bars between haplotypes represent the number of nucleotide differences between each haplotype. The small red circles indicate hypothetical haplotypes samples on PubMLST, for which only the *gatB* gene for POW471 had an exact match. The other sequences were based on the closest related matches from the PubMLST blast results. The *coxA* sequences had the most similarity between what could be tested for each of the samples. Only one gene sequence had an exact match, suggesting that the other strains could be new sequence types that have not been documented previously.

**Figure 5 biology-13-00121-f005:**
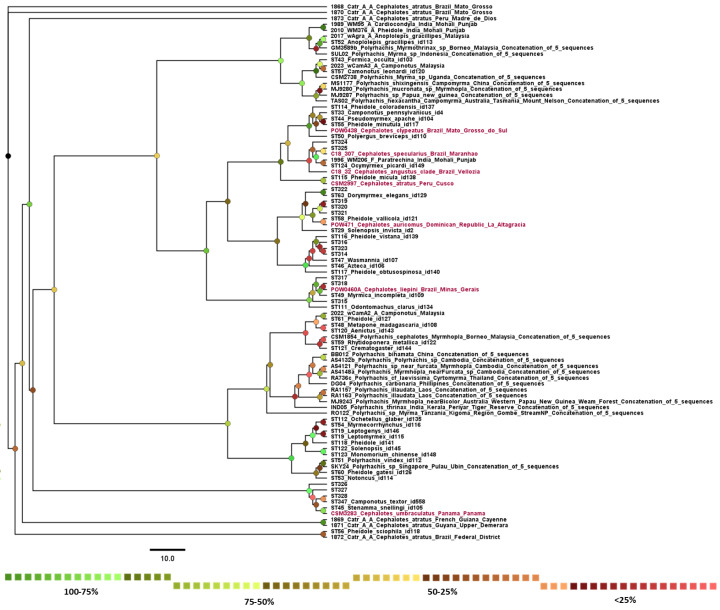
Phylogram representing aligned MLST sequences from *Wolbachia* in ant hosts. Sample names in red font indicate hosts of new sequences used for this study. Bootstrap support is indicated by labeled nodes; colors correspond to the percentage ranges indicated in the legend on the right.

**Table 1 biology-13-00121-t001:** The species, location, and collector information for the positive samples. The first column states what infection type they had (single or multiple). A total of 18 samples were positive.

Species ID	Infection	Species	Country	Species ID	Infection	Species	Country
C18_208	Multiple	*Cephalotes nilpiei*	Brazil	CSRA1209a	Multiple	*Cephalotes pallens*	Brazil
**C18_25**	Multiple	*Cephalotes pinelii*	Brazil	JSC203D	Multiple	*Cephalotes goniodontus*	Mexico
**C18_307**	Single	*Cephalotes specularis*	Brazil	POW0403	Multiple	*Cephalotes pallidoides*	Brazil
**C18_32**	Single	*Cephalotes angustrus* clade	Brazil	POW438	Single	*Cephalotes clypeatus*	Brazil
**CSM2997**	Single	*Cephalotes atratus*	Peru	POW460A	Single	*Cephalotes liepini*	Brazil
**CSM3360**	Multiple	*Cephalotes umbraculatus*	Panama	POW471	Single	*Cephalotes auricomus*	Dominican Republic
**CSM3433**	Single	*Cephalotes cordatus*	French Guiana	POW547A	Multiple	*Cephalotes angustus*	Brazil
**CSM3283**	Single	*Cephalotes umbraculatus*	French Guiana	SP81a	Multiple	*Cephalotes placidus*	Peru
**CSRA1185**	Single	*Cephalotes cristatus*	Costa Rica	SP83a	Multiple	*Cephalotes clypeatus*	Peru

**Table 2 biology-13-00121-t002:** The allele and *Cephalotes* host information is shown for the seven positive samples with at least one successfully sequenced MLST gene. One of the samples, CSRA_1185, did not produce any MLST alleles, so it was removed from the table. Two dashes (--) indicate that a sequence was not able to be produced from that sample for the specific gene, or in the case of the sequence type, not enough alleles were available to make a determination. A blue-colored box indicates that the number is the closest associated allele and or sequence type determination. The country of origin and species determination are located in the last two columns.

	MLST Allele Number							
Sample ID	coxA	fbpA	ftsZ	gatB	hcpA	Sequence Type	Country	Host Species
C18_32	20	253	43	196	45	315	Brazil	*C. angustrus*
C18_307	--	--	--	--	45	--	Brazil	*C. fiebrigi*
CSM2997	--	20	--	--	--	--	Peru	*C. atratus*
CSM3283	20	--	37	--	44	--	French Guiana	*C. umbraculatus*
CSM3433	--	--	--	--	45	--	French Guiana	*C. cordatus*
POW0438	299	45	--	--	55	--	Brazil	*C. clypeatus*
POW0460A	20	46	--	19	55	--	Brazil	*C. pinelii*
POW471	20	253	--	19	305	--	Dominican Republic	*C. hamulus*

## Data Availability

New *Wolbachia* sequences are available on the National Library of Medicine’s National Center for Biotechnology Information GenBank at accession PP297504-PP297525.

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
