# Peer review of "The Diversity of Wolbachia across the Turtle Ants (Formicidae: Cephalotes spp.)"

_biology, 2024, doi:10.3390/biology13020121_

Round 1
Reviewer 1 Report
Comments and Suggestions for Authors
Dear colleagues,
your research of Wolbachia distribution in Cephalotes ants is very informative and perspective in study of intestinal microorganism function within invertebrates. This trend seems to be important in further study of symbiont impact on the host organism, which may provide unique characters such as ability of wood or other cellulose-containing substrates destruction. Your study is quite informative, but further widening of the geographic area of the host species distribution and Cephalotes species-specific connections with Wolbachia would be necessary to get more clear picture of microbe-ant interaction.
I am pleased recommending your study for publication as it is, but it seems necessary revise the text to make it much more correspond to the Rules of the Journal. Also, some tables or figures (as Table 1, for example) are very difficult to read and understand, but I have no idea how to rectify them.
Author Response
Thank you for your feedback on our study. We agree with you that it is necessary to widen the geographic area of the Cephalotes species distribution to understand the Wolbachia-Cephalotes interaction better; we hope to do so in future work. We have made changes to some tables and figures to improve readability. Thank you again for your edits and we hope that our changes have improved the manuscript greatly.
Reviewer 2 Report
Comments and Suggestions for Authors
The manuscript presented by the excellent research group presents important information about the Wolbachia x turtle ants relationship that deserves to be published in the journal Biology. Before acceptance, I suggest small modifications:
1. I suggest that authors include the name(s) of the author(s) and year of publication every time the genus or species is mentioned for the first time in the text.
2. Line 31: change “wolbachia” for “Wolbachia”
3. Line 18: change “Baldo et al. (2006)” for “Baldo et al.18”
4. Line 70 and 71: if "16S rDNA, wsp, and ftsZ markers" are related to genes, they need to be presented in italics
5. Line 72: change “Ramalho and Moreau (2020)” for “Ramalho and Moreau22”
6. Line 98: change “Cephalotes atratus” for “C. atratus”
7. Lines 114-117: check the spacing between words
8. Line 126: change “Wolbachia” for “Wolbachia”
9, Line 144: Table 1 is not in the magazine's standards. Furthermore, it is not possible to read its data. Authors need to transform the image into a table (which can be presented horizontally with an appropriate font size for readers to understand).
10. Line 148: change “Multilocus Sequence Typing (MLST)” for “MLST”
11. Line 150: change “wsp gene” for “wsp gene”
12. Line 163: change “Paratrechina sp.” for “Paratrechina sp.”
13. Line 191: change “Cephlotes cristatus” for “C. cristatus”
14. Line 210: table 2: change the scientific names to: C. angustrus, C. fiebrigi...
15. Line 215: the quality of figure 5 needs to be improved and the size needs to be enlarged to better read the information.
16. Line 218: the quality of figure 6 needs to be improved and the size needs to be enlarged to better read the information. In the form presented, it is not possible to read any scientific name.
18 Line 150: change “Cephlotes atratus” for “C. atratus”
Author Response
Thank you for your feedback. We agree with all 18 modifications you have suggested and have implemented them in our updated manuscript. We hope that our changes in response to your edits have greatly improved our manuscript.
Reviewer 3 Report
Comments and Suggestions for Authors
The language of the whole article makes it very difficult for the reader to read, leading to great difficulties in understanding the article.
1. The existence of Figures 1 and 2 seems unnecessary.
2. The purpose and significance of gut microbiota research were not clearly written in the article.
Comments on the Quality of English LanguageLanguage repetition of repetitive, need to be concise and refined. Writing to use short sentences as far as possible, make it easier for readers to read.
Author Response
Thank you for your feedback. We have modified the language in the manuscript, which will hopefully make it easier to read. We have decided to keep Figures 1 and 2, though we have modified Figure 2. If you feel strongly that these figures are unnecessary we can move them to the supplemental materials. We have also changed the results and conclusions to make the significance of the gut microbiota research more clear.
Reviewer 4 Report
Comments and Suggestions for Authors
Reese et al sample 116 ant samples from across Central and South America to determine the presence and diversity of Wolbachia. For those samples positive for a single strain of Wolbachia, they further type them by sequencing of MLST loci and compare their results to existing sequences in the MLST database. Overall, they find many unique Wolbachia alleles for most of the loci. The authors also test and do not find any correlations between the Wolbachia phylogeny and host phylogeny nor the location of sample collections. Overall, this study adds to the knowledge of Wolbachia prevalence and diversity in Cephalotes spp. I have recommended that the authors provide a bit more context for some of the work they have done for more general readers and alter many of the figures and tables to make them more clear and accurate. Additionally, I do think the authors need to be a bit careful about some of their conclusions given their inability to sequences most MLST loci across their samples. See my recommendations below.
Figure 1: The authors should include a legend to indicate the ranges of samples that the colouring indicate.
Methods:
Line 123: Were samples stored dry at -20C or in ethanol or some other preservative?
Line 127: Cite the publication for these primers. Also cite the publication containing sequences for the MLST primers.
Table 1 is hard to read. Since most of this data is also included in the supplement, can some less important data be included here to allow for larger font size? Also, there is a question mark after meters in the elevation column. Confirm the data is in meters.
Figure 2: I would recommend removing the total samples column. It doesn’t add any information and the x-axis label (infectious status) is not reflected in this class.
Figure 3: I’m confused what A is showing. The label says distribution of all the samples that were tested. Isn’t that shown in Figure 1 already? Also the legend indicates that the max is 8 samples, but your supplemental tables shows there are more than 8 samples from Brazil. Additionally, it is quite difficult to see some of the positive countries since the whole globe is included. I would recommend to limit the figures to the western hemisphere as was done for Figure 1 to allow for larger images.
Figure 4: Enlarge x-axis labels to make them more legible.
Table 2: In the legend and/or the main text, the authors should clarify or provide a bit of context regarding what they mean by allele number for readers who aren’t familiar with the pubMLST database. E.g. MLST sequences were blasted against the pubMLST database to identify close matches and that alleles present in the database are numbered as most people would likely expect such a search to produce a strain that is the most closely related.
The authors mention that 9 samples were singly infected (Table 1 confirms). Sample CSRA1185 did not amplify any MLST loci and was excluded. However, only 7 samples are included in Table 2. CSM3433 seems to have been lost from Table 2, while it is included for one MLST locus in Figure 5?
Figure 5: Indicate which C18 samples are referred to for ftsZ and gatB. There is an additional haplotype (CSM 3433) included in hcpA that isn’t included anywhere else nor is it listed in Table 2.
Paragraph starting at line 230: Figure references are incorrect.
Figure 6: The figure legend does not encompass all the colours in the phylogeny.
Conclusions:
The authors have interpreted their phylogenetic analysis showing Wolbachia from the same species across multiple clades as potentially wide biodiversity of Wolbachia within ant genera. Given that the authors were only able to amplify one MLST gene for CSM2997, it seems quite likely that there simply isn’t enough data to confidently place these sequences and a caveat should be stated regarding this conclusion. How many MLST genes are included in the phylogeny for the other C. atratus sample?
Minor
Line 67: extra comma
Line 224: sentence fragment. I think here is where you could add a bit more detail regarding the MLST database that I mentioned above.
Gene names should be italicised.
Results paragraph starting at line 238: This is a stylistic preference that authors can decide on: I think it would be ideal for a reader if the authors provided a short context as to why they were performing the mantel tests. I realise that information is in the methods, but those are often not read or quickly forgotten.
Author Response
Thank you for your feedback. We have incorporated much of what you suggested. Below is a point-by-point response to each edit:
Figure 1: The authors should include a legend to indicate the ranges of samples that the colouring indicate. DONE
We have added a legend to Figure 1.
Methods:
Line 123: Were samples stored dry at -20C or in ethanol or some other preservative?
The samples were collected and stored in ethanol immediately and at -20C after return from the field. This detail has been added to the manuscript.
Line 127: Cite the publication for these primers. Also cite the publication containing sequences for the MLST primers.
Both the wsp and MLST primers were designed by and published on in Baldo et al., 2006. The citation has been added to Line 127.
Table 1 is hard to read. Since most of this data is also included in the supplement, can some less important data be included here to allow for larger font size? Also, there is a question mark after meters in the elevation column. Confirm the data is in meters.
We have excluded some data to allow for a larger font size and corrected the mistake.
Figure 2: I would recommend removing the total samples column. It doesn’t add any information and the x-axis label (infectious status) is not reflected in this class.
We actually removed this entire figure because of feedback from other reviewers.
Figure 3: I’m confused what A is showing. The label says distribution of all the samples that were tested. Isn’t that shown in Figure 1 already? Also the legend indicates that the max is 8 samples, but your supplemental tables shows there are more than 8 samples from Brazil. Additionally, it is quite difficult to see some of the positive countries since the whole globe is included. I would recommend to limit the figures to the western hemisphere as was done for Figure 1 to allow for larger images.
Figure 2 is now a modified version of the old Figure 3, and we have taken your suggested feedback.
Figure 4: Enlarge x-axis labels to make them more legible.
We have enlarged the x-axis labels.
Table 2: In the legend and/or the main text, the authors should clarify or provide a bit of context regarding what they mean by allele number for readers who aren’t familiar with the pubMLST database. E.g. MLST sequences were blasted against the pubMLST database to identify close matches and that alleles present in the database are numbered as most people would likely expect such a search to produce a strain that is the most closely related.
We have clarified allele number to hopefully make it easier to understand.
The authors mention that 9 samples were singly infected (Table 1 confirms). Sample CSRA1185 did not amplify any MLST loci and was excluded. However, only 7 samples are included in Table 2. CSM3433 seems to have been lost from Table 2, while it is included for one MLST locus in Figure 5?
Thank you for catching this! CSM3433 was excluded from Table 2 in error and has been added back in.
Figure 5: Indicate which C18 samples are referred to for ftsZ and gatB. There is an additional haplotype (CSM 3433) included in hcpA that isn’t included anywhere else nor is it listed in Table 2.
The samples for ftsZ and gatB have been indicated. CSM3433 is now included in Table 2.
Paragraph starting at line 230: Figure references are incorrect.
This has been fixed.
Figure 6: The figure legend does not encompass all the colours in the phylogeny.
Figure 6 has been changed and the new figure legend should encompass the colors in the phylogeny.
Conclusions:
The authors have interpreted their phylogenetic analysis showing Wolbachia from the same species across multiple clades as potentially wide biodiversity of Wolbachia within ant genera. Given that the authors were only able to amplify one MLST gene for CSM2997, it seems quite likely that there simply isn’t enough data to confidently place these sequences and a caveat should be stated regarding this conclusion. How many MLST genes are included in the phylogeny for the other C. atratus sample?
We agree with you and this caveat has been added in the discussion. The other C. atratus samples (sequenced by Kelly et al) have all MLST genes sequenced.
Minor
Line 67: extra comma DONE
We have corrected this typo.
Line 224: sentence fragment. I think here is where you could add a bit more detail regarding the MLST database that I mentioned above.
We have added more information and corrected the sentence fragment.
Gene names should be italicised.
We have italicized the gene names.
Results paragraph starting at line 238: This is a stylistic preference that authors can decide on: I think it would be ideal for a reader if the authors provided a short context as to why they were performing the mantel tests. I realise that information is in the methods, but those are often not read or quickly forgotten.
We have added this information. Thank you for the suggestion.
Round 2
Reviewer 4 Report
Comments and Suggestions for Authors
I am satisfied with the authors' changes. Thank you for your careful consideration.